# Phosphate availability and implications for life on ocean worlds

Noah G. Randolph-Flagg ●[1,2,3] ✉, Tucker Ely[2,4,5], Sanjoy M. Som[1,3], Everett L. Shock[5], Christopher R. German ●[6] & Tori M. Hoehler ●[1]

Several moons in the outer solar system host liquid water oceans. A key next step in assessing the habitability of these ocean worlds is to determine whether life's elemental and energy requirements are also met. Phosphorus is required by all known life and is often limited to biological productivity in Earth's oceans. This raises the possibility that its availability may limit the abundance or productivity of Earth-like life on ocean worlds. To address this potential problem, here we calculate the equilibrium dissolved phosphate concentrations associated with the reaction of water and rocks−a key driver of ocean chemical evolution−across a broad range of compositional inputs and reaction conditions. Equilibrium dissolved phosphate concentrations range from $10^{-11}$ to $10^{-1}$ mol/kg across the full range of carbonaceous chondrite compositions and reaction conditions considered, but are generally $> 10^{-5}$ mol/kg for most plausible scenarios. Relative to the phosphate requirements and uptake kinetics of microorganisms in Earth's oceans, such concentrations would be sufficient to support initially rapid cell growth and construction of global ocean cell populations larger than those observed in Earth's deep oceans.

The search for life in the solar system has focused on the presence of liquid water, the availability of nutrients required for life on Earth, and the chemical energy required for metabolism[1]. Beneath insulating ice shells, global oceans have been proposed to exist on Jupiter's moons Europa, Ganymede, and Callisto; Saturn's moons Titan, Enceladus, Mimas, and Dione; Neptune's moon Triton; Uranus's moons Ariel and Miranda; and the dwarf planets Ceres, Pluto and Pluto's moon Charon[2]. Spacecraft observations indicate potentially habitable conditions on both Europa, the target of the forthcoming "Europa Clipper" mission, and Enceladus, which has been prioritized for study by the NASEM Decadal Survey on Planetary Science and Astrobiology[3].

On Enceladus, plumes of water ice and vapor emerge from fractures at the south pole[4]. The Cassini spacecraft repeatedly flew through these erupted plumes and was able to characterize the chemistry of the source fluid sufficiently to suggest favorable conditions for life as we know it on Earth. The temperatures and pressures of the ocean lie within the range observed in Earth's oceans[5], the pH appears to be somewhat basic (pH 8–11)[6,7], comparable to estimates of the early Earth ocean (<9)[8,9], and the range of ocean salinity inferred from particles in Saturn's E-ring (0.05–0.2 mol/kg)[10] falls well within the range that is habitable to Earth organisms. Silica nanoparticles are interpreted to indicate active hydrothermal systems (vent fluids >50 °C)[11] and observations of co-occurring $CO_2$ and $H_2$ indicate the presence of chemical disequilibrium that could support methanogenic life[12]. The 'biogenic elements' carbon, hydrogen, nitrogen, and oxygen are all abundant relative to life's requirements[6].

For Europa, magnetometer observations made by the Galileo spacecraft are interpreted as indicating the presence of a saline ocean that has reacted extensively with an underlying silicate crust[13,14], potentially providing both material resources and energy for life. Near-infrared spectroscopy indicates the presence of radiolytically-produced oxidants at the surface of Europa's ice shell that, if

[1]Space Science and Astrobiology Division, NASA Ames Research Center, Moffett Field, Mountain View, CA, USA. [2]NASA Postdoctoral Program, Universities Space Research Association, Columbia, MD, USA. [3]Blue Marble Space Institute of Science, Seattle, WA, USA. [4]Department of Earth and Environmental Sciences, University of Minnesota, Minneapolis, MN, USA. [5]School of Earth and Space Exploration, Arizona State University, Tempe, AZ, USA. [6]Dept. Geology and Geophysics, Woods Hole Oceanographic Institution Woods Hole, Falmouth, MA, USA. ✉e-mail: nrflagg@berkeley.edu

delivered to the ocean via ice circulation, could represent abundant energy for life when reacted with geochemically-sourced reductants[15].

The availability of phosphate, which is limiting to biological productivity in portions of Earth's oceans and is thought to be limiting to life over geologic time[16], has now become a point of focus. Phosphorus is a necessary nutrient for all known life on Earth where, in the 5+ oxidation state (equivalent to orthophosphate), it forms part of both the information-encoding molecules DNA and RNA and the key energy-carrying molecule ATP. It is theorized that phosphate may be required for information-carrying molecules in *any* form of life[17] due to its ability to form multiple bonds while retaining charge[18]. In many parts of Earth's oceans, cell growth rates, population dynamics, and overall biological productivity are determined by phosphate availability[19]. With these points of reference, recent work has begun to explore whether phosphate availability might be limiting to life beyond Earth[20–22].

On the ocean worlds of our solar system, as on Earth, phosphate availability is likely not a question of bulk phosphorus abundance but, rather, of the processes that govern its partition into the ocean. The bulk inventory of phosphorus in the silicate crustal materials of ocean worlds is likely vast relative to even a large biosphere's requirements. Phosphorus abundance is consistently near 0.1 weight % across the diversity of carbonaceous chondrites[23] ($n = 1800$, [P] = 0.11 ± 0.02 wt%; Fig. 1a), the materials from which the solid bodies of the solar system are thought to have formed. At such levels, Cable and co-workers[24] calculated that the phosphorus required to establish a global ocean cell abundance on Enceladus of $10^6$ cells/mL (comparable to Earth's deep oceans[25]) is present in just the upper few centimeters of the silicate crust, assuming a per-cell phosphorus requirement equivalent to that of aquatic cells on Earth. On large bodies, the differentiation of silicate crust into mineralogically distinct reservoirs could lead to the depletion of phosphorus in the upper crust if it preferentially partitions into deeper reservoirs. However, this effect appears to be minor: the mean phosphorus abundance in mid-ocean ridge basalts on Earth[26] ($n = 3598$, [P] = 0.074 ± 0.04 wt%), which results from such differentiation, differs only slightly from that of carbonaceous chondrites (Fig. 1a). Thus, whether phosphate is limiting to life in the ocean worlds of our solar system is not a question of bulk phosphorus abundance but rather of the processes that govern its partitioning into the ocean from the vast reservoir that is represented in the silicate interiors of those worlds.

Efforts to model the processes that liberate phosphate in, and sequester it from, the ocean have been made for Earth and, increasingly, worlds beyond Earth. An extensive body of work has been devoted to characterizing and constraining the processes that govern phosphorus availability in Earth's oceans, both now and over geological timescales[16,27–30]. Presently, the weathering of granitic-composition continental crust is the dominant source of phosphate to the oceans[31], and hydrothermal systems are thought to be the main

abiotic sink[32]. In extending these observations to Enceladus and Europa, Lingam and Loeb[20] observed that neither the crustal processing required to produce granite nor the rainfall and runoff required to weather granite can exist on small, ice-covered satellites. By contrast, geochemical evidence[11,33] and theoretical models[34,35] suggest that hydrothermal circulation (a putative sink for phosphate) may be active and have persisted through geologic time on Enceladus and Europa[36,37]. By considering the effect of fluid pH on silicate dissolution rates, Lingam and Loeb[20] concluded that an alkaline ocean pH on Enceladus could support only a very slow release of phosphate by the dissolution of primary minerals, such that the relative balance of sinks and sources would result in severe phosphate limitation. Conversely, Lingam and Loeb[20] postulated that putatively acidic conditions on Europa would yield higher silicate dissolution rates and, consequently, a phosphate-replete ocean.

Here, we consider that the processes that liberate phosphate from primary minerals and sequester it by incorporation into secondary minerals are governed by chemical equilibria. Consequently, a 'sink' does not consume phosphate to extinction but rather to a non-zero equilibrium concentration that is governed by a range of compositional and environmental variables.

Measurements made in hydrothermal systems on Earth, while limited, demonstrate that phosphate remains relatively abundant in venting fluids. The most exhaustive published database of submarine hydrothermal vent chemistry[38] contains 2388 vent fluid compositions but only 43 of those, representing four vent fields, have associated measurements of phosphate. Dissolved phosphate concentrations at these sites, which span temperatures of 50–350 °C, range from 3.2 to $33.3 \times 10^{-7}$ mol/kg (Fig. 1b). Surveys at individual sites suggest that phosphate concentrations systematically decrease by a factor of 10 at higher temperature[32]. Similarly, iron-rich hydrothermal plume minerals in oxygenated seawater can further scavenge phosphate[16,39], lowering water column concentrations within the region of the plume by several hundred nanomoles/kg, relative to deep ocean concentrations (e.g., from $2.6 \times 10^{-6}$ to $2.3 \times 10^{-6}$ mol/kg)[40]. Both processes represent 'sinks' only relative to deep ocean phosphate concentrations that are enriched by the degradation of sinking biological material. Relative to a phosphorus-poor ocean, hydrothermal vent fluids could instead serve as significant sources of phosphate. Indeed, phosphate is actively taken up by biology in many parts of Earth's oceans that have phosphate concentrations three orders of magnitude lower than the minimum values reported at these hydrothermal vents[19], demonstrating that the seemingly low concentrations associated with hydrothermal processes are nevertheless ample from a biological perspective. Hao and and co-workers[20] have recently shown that even higher phosphate concentrations ($10^{-7}$ to $10^{-2}$ mol/kg $H_2O$) are implied by equilibrium with respect to the inferred temperature (0 °C), pH, dissolved inorganic

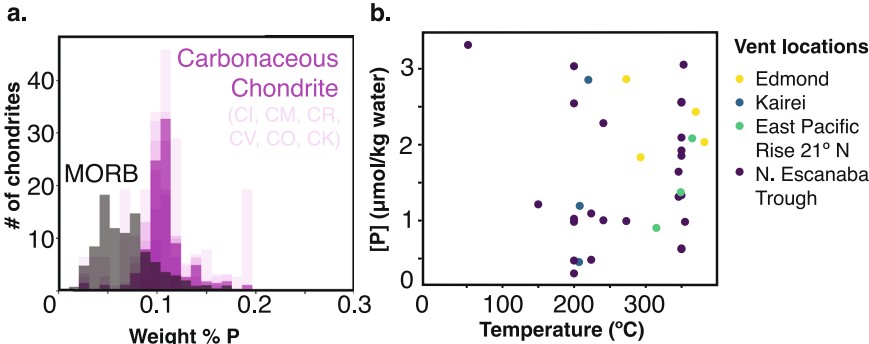

**Fig. 1 | Phosphorus abundances in chondrites, basalts, and hydrothermal fluids. a** Histogram of phosphorus concentrations (wt%) for all carbonaceous chondrite compositions in the chondrite database[23] ($n = 1800$) and Mid Ocean Ridge Basalt database[26] ($n = 3598$). All chondrite compositions are depicted at 10% opacity to highlight the large degree of overlap among different chondrite classes. **b** Phosphorus concentrations from the submarine hydrothermal vent database ($n = 34$)[38].

content, and redox state of Enceladus' ocean, and concluded that any life in that ocean should not be inhibited by low phosphate availability.

Ultimately, both equilibrium processes[22] and dynamical processes[20] could represent important controls on phosphate availability in ocean worlds. To encompass both considerations, here we model the equilibrium phosphate concentrations established during aqueous alteration of a range of chondritic compositions, at a broad range of reaction conditions, with two objectives. The first is to encompass the possibility that bulk ocean chemistry can be strongly influenced by equilibria established at conditions that lie far from bulk ocean conditions. As a simple example, magnesium concentrations in Earth's oceans are set in large part by sequestration that occurs during hydrothermal circulation[41], representing much higher temperatures, much more reducing conditions, and lower pH than in Earth's bulk oceans. For the specific case of Enceladus, our focus on the range of hydrothermal conditions and potential dynamic controls on ocean phosphate abundance provides a distinct and complementary perspective in relation to Hao and co-workers' focus on equilibrium with respect to bulk ocean conditions. The second objective is to provide a broad basis for assessing the question of phosphate availability across multiple ocean worlds that could span a range of compositional inputs and reaction conditions. We explore the biological implications of our results through reference to the phosphate requirements and uptake dynamics of modern aquatic cells on Earth. We do not explore the implications for origin-of-life chemistry because the requirements of that chemistry are much less well-constrained[42]. However, the results of this study can inform future efforts to do so.

## Results
### Equilibrium dissolved phosphate concentrations
We calculated the equilibrium aqueous phosphorus concentrations resulting from reaction of chondritic rocks with water via the geochemical modeling software, EQ3/6[43], employing mineral- and species-specific thermodynamic data[44,45]. EQ3/6 computes the fluid chemistry and mineral assemblage that would result from reacting specified quantities (masses) of water and rock to equilibrium at a specified temperature and pressure. The calculations do not incorporate reaction kinetics and provide no information about timescales of reaction, although recent experimental studies[22,46–48] suggest that dissolution of phosphate minerals is rapid on geological timescales over a wide range of conditions. Instead, the extent of reaction is represented in our calculations by the water-to-rock mass ratio (hereafter "W:R"), with high and low ratios reflecting limited and extensive reaction, respectively.

We focus specifically on the role of alteration mineral phases in setting equilibrium dissolved phosphorus concentrations. To isolate these controls, our simulations reacted chondritic materials with "minimal" fluid (neutral, anoxic water with initial element concentrations each set to $10^{-15}$ mol/kg) and did not encompass organic-phosphate chemistry[49] or transiently stable phases such as phosphides[42]. Here, "chondrite" refers specifically to the carbonaceous chondrite class of stony meteorites, which are thought to be the most primitive solar system material[50]. While the rocky cores of ice-covered ocean worlds are not directly observable, the bulk compositions of most planetary bodies are consistent with the elemental abundances in such chondrites. As a result, models of ocean world chemical evolution commonly assume a chondrite-like initial composition[6,51]. Our calculations follow suit, and encompass the variability observed across six major chondrite classes[50] with respect to their inventory of Al, C, Ca, Cl, F, Fe, H, K, Li, Mg, Mn, Na, O, P, S, and Si (Supplemental Table 1)[23], while considering the equilibria associated with a range of phosphate and phosphite species (Supplemental Table 2). Reactions were simulated at temperatures ranging from 1 to 300 °C and W:R ranging from 0.1 to $10^6$. All calculations were performed at 50 MPa, which is reflective of the pressures inferred for water rock reactions within proposed hydrothermal aquifers on Enceladus and

Europa[5,35], although the water-rock reactions that impact phosphate concentrations appear relatively insensitive to pressure[22].

In all simulations, water-rock reactions are a source of dissolved phosphorus relative to the initial "minimal" fluid composition. Orthophosphate species (e.g., $H_2PO_4^-$, $HPO_4^{2-}$, $PO_4^{3-}$) accounted for >99.9% of dissolved phosphorus in all simulations, consistent with the findings of Hao and co-workers[20] for Enceladus ocean conditions. We hereafter refer to the sum of the concentrations of these species as "dissolved phosphate" and focus the subsequent discussion exclusively on this pool.

Elemental compositions vary across the six main carbonaceous chondrite types[50]. We assessed the impact of this variability on dissolved phosphate concentrations with a separate suite of calculations applied to each type. Across the range of chondrite types considered, the distribution of total dissolved phosphate follows a similar pattern with respect to temperature and W:R, with CI chondrite producing the most phosphate (Fig. 2). Among the six chondrite types considered, CI chondrites—which consistently yielded higher dissolved phosphate in our simulations—most closely match solar system element abundance ratios and are generally considered as most representative of the source material for solar system formation[6,51]. Hence, we focused on CI chondrites as the basis for a more detailed analysis.

### Effects of redox conditions
CI chondrites contain abundant sulfur, iron, and carbon[52] and the oxidation states of these elements are known to vary among chondrites[53]. Moreover, in the specific case of Enceladus, it has been proposed that the silicate interior may comprise a carbonate-rich upper layer and a serpentinizing deeper layer, with potential to expose circulating fluids to a diverse mineralogy that encompasses varying oxidation states[54]. To assess the sensitivity of equilibrium dissolved phosphate concentrations to the redox state of the reacting materials, we performed a suite of calculations in which the oxygen mass fraction was varied over a range that is equivalent to fixing the sulfur and iron oxidation states at 2- and 2+, respectively, and varying the carbon oxidation state from zero to 4+. Given the large mass fraction of carbon in chondrites (~3.5%, Supplemental Table 1), this approach explores a large range of bulk average redox states in the reacted material. The subsequent discussion is based on an average carbon oxidation state of 2+, such that our sensitivity analysis ranges to conditions that are both more reducing (Fig. 3a) and more oxidizing (Fig. 3c) relative to that nominal case (Fig. 3b). Broadly, a shift to more oxidizing conditions has little effect, with dissolved phosphate comparably abundant or somewhat higher (Fig. 3c and Supplemental Fig. 1). A more reduced composition in the reacting solid phase yields phosphate concentrations 2–4 orders of magnitude lower within the middle range of water-to-rock ratio but nevertheless maintains a core of high (>$10^{-4}$ mol/kg) phosphate at low to moderate reaction temperatures and low W:R (Fig. 3a and Supplemental Fig. 1).

In all simulations, hydroxyapatite, a calcium phosphate mineral, is the dominant phosphate sink among the minerals and aqueous complexes considered in these calculations. Compositional inputs, reaction conditions, and mineral equilibria that yield higher dissolved calcium concentrations and/or decrease the solubility of hydroxyapatite will lead to correspondingly lower dissolved phosphate concentrations. This control underlies the variations with temperature and W:R that occur for a fixed compositional input and also the differences observed among chondrite types (Supplemental Fig. 1d–i). For example, the higher dissolved phosphate that occurs with a more oxidizing parameterization of CI chondrite composition is due primarily to the more pervasive precipitation of carbonates in those cases (Fig. 3d–i), which decreases the availability of Ca for hydroxyapatite precipitation.

### Application to other ocean worlds
By virtue of our focus on chondritic materials, the results are most directly applicable to small bodies (e.g., Enceladus, the small moons of

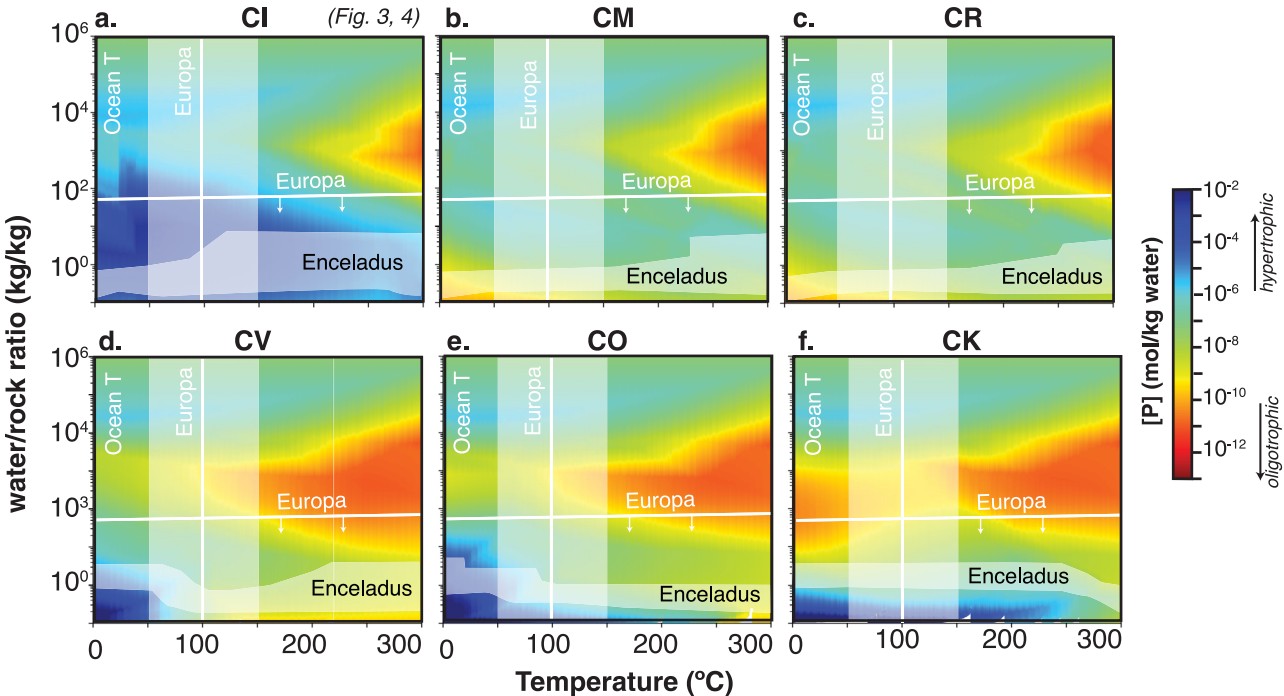

**Fig. 2 | Total dissolved phosphate concentration (mol/kg) for averaged carbonaceous chondrite composition for different chondrite classes and mid-ocean ridge basalts with 0–2 wt% C. a–c** CI, CM, CR are closer to solar stoichiometry and **d–f** CV, CO, CK have higher Ca concentrations. For comparison with Europa, vertical lines are inferred from hydrothermal models[35] and the horizontal line is the lower bound on water/rock ratio inferred from salinity estimates[13]. For comparison with Enceladus, shaded regions show estimates from Si nanoparticles[11] while horizontal lines track modeled [Na] corresponding to observations of NaCl observed in the E-ring[10] (additional pH constraints Fig. 3).

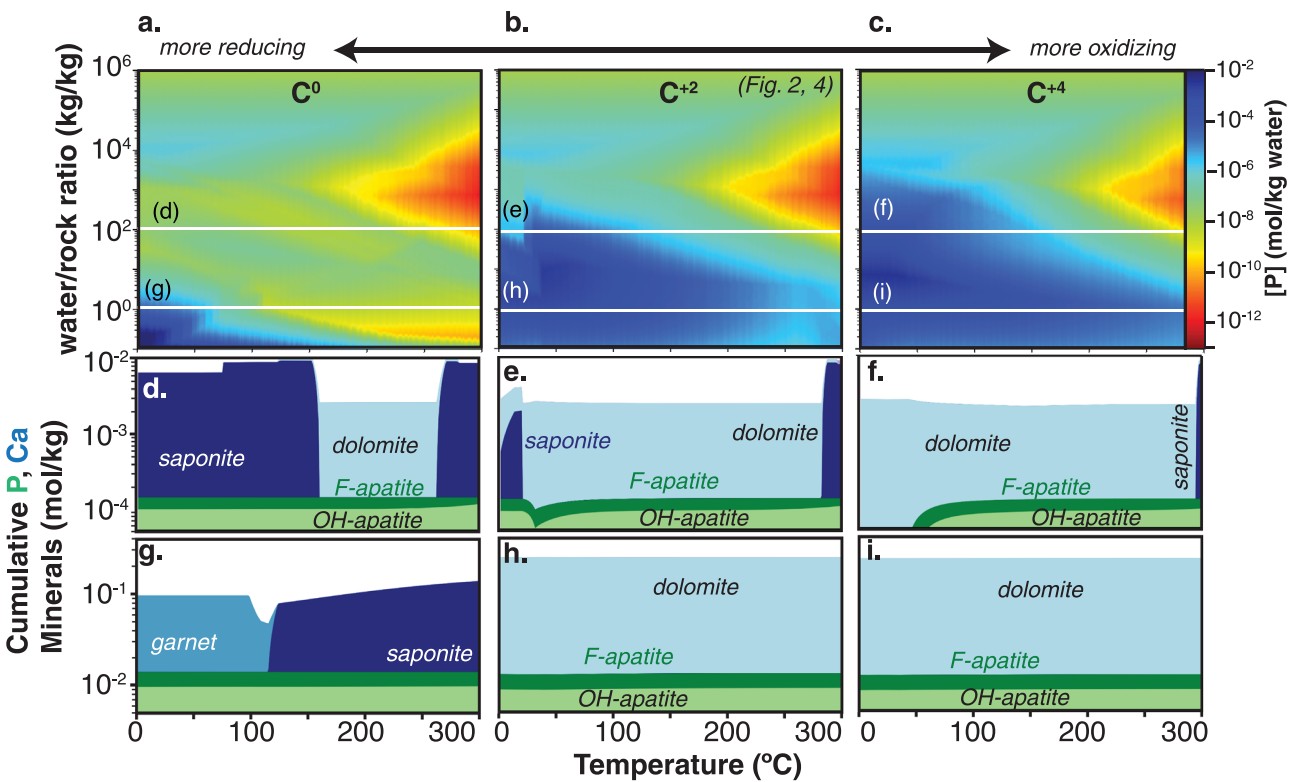

**Fig. 3 | Dissolved phosphate and phosphate and carbonate mineral deposition at range of water-rock ratios and temperatures. a–c** Total dissolved phosphate concentration (mol/kg) for averaged CI carbonaceous chondrite composition for different redox states of carbon. Ca and P bearing minerals for water/rock ratio of 100 **d–f** and 1 **g–i.**

Uranus, and other bodies) that are unlikely to have experienced extensive silicate melting and differentiation. On larger bodies, potentially including Europa[37,55], magmatic processing that impacts the bulk chemical inventory and mineralogy of crustal rocks has an associated potential to impact the dissolved phosphate concentrations that result from fluid-rock interactions. Preliminary calculations indicate that basaltic mineralogy yields lower dissolved phosphate concentrations (~$10^{-7}$ to $10^{-9}$ mol kg$^{-1}$) when reacted with "minimal fluid" (Supplemental Fig. 2a). However, magmatic crustal processing encompasses additional chemical evolution within the system that could yield significantly higher concentrations. For example, when the calculations include the volatiles produced during magmatism (e.g., $CO_2$[26]), dissolved phosphate increases by several orders of magnitude (~$10^{-5}$ to $10^{-2}$ mol kg$^{-1}$ for W:R < 1000; Supplemental Fig. 2b). Similarly, the proposed metamorphic origin of Europa[55] encompasses significant precipitation of gypsum ($CaSO_4$) that, by depleting the $Ca^{2+}$ required for apatite precipitation, could yield phosphate concentrations that are elevated relative to the MORB-only calculations. Future work that considers such system-level chemical evolution effects will therefore be valuable for predicting the dynamics of dissolved phosphate on larger bodies. We do not consider our results directly applicable to cases, e.g., Ganymede and Titan, where the high pressures and chemical phases involved extend well beyond the conditions we modeled; prior work indicates that the attendant phosphorus chemistry in such cases may differ significantly from that considered here[56].

## Discussion

Ultimately, it is the bulk ocean abundance of phosphate that is most relevant in considering the question of phosphate availability to life on ocean worlds. However, that abundance could be dynamically controlled, to varying extents, by reactions that occur as ocean fluids circulate through the silicate crust at conditions different from those in the ocean[20]. By spanning a broad range of compositions and conditions, our model results bear on both bulk ocean evolution and the water-rock chemistry that may occur during fluid circulation, and thus enable us to consider the implications of dynamic control on bulk ocean phosphate.

Bulk ocean chemical evolution integrates across a range of reactions that may vary widely with respect to both compositional inputs and reaction conditions[57]. Despite the potential for such heterogeneity, observational constraints on the extent of water-rock reaction, as W:R, suggest that most plausible fluid inputs into the ocean have relatively abundant phosphate (Fig. 2). In the context of ocean chemical evolution, the transition from high to low W:R represents the accumulation over time of water-rock reaction products, as ocean fluids circulate through the silicate crust, react with the rocks, and reemerge into the ocean in altered form[6,51]. Observational data constrain W:R for both Enceladus and, to a lesser extent, Europa. Cassini observations of NaCl in materials from Saturn's E-ring, which are thought to be sourced from the plumes of Enceladus, are interpreted as representing a bulk ocean concentration of 0.05-0.2 mol/kg NaCl[10]. In our simulations, which track the evolution of fluid $Na^+$ concentrations, the observed range of [Na] corresponds to W:R in the range of 0.1 to 6 (Fig. 2) – similar to the range (0.3-1) suggested by Glein and Waite (2020) for the ocean of Enceladus. Within the W:R range of 0.1 to 6, equilibrium phosphate concentrations are consistently > $10^{-5}$ mol/kg for CI chondrite composition. For Europa, Galileo magnetometer data are interpreted as placing a lower bound on ocean salinity that corresponds to W:R < 40[58], corresponding to phosphate concentrations consistently > $10^{-8}$ mol/kg for CI chondrite composition. It is likely that bulk ocean phosphate concentrations would significantly exceed these lower bounds. In all simulations, the lowest equilibrium phosphate concentrations are associated with high reaction temperatures and/or compositions corresponding to the less common chondrite classes (Fig. 3). As such, low bulk ocean phosphate concentrations would only

be possible in a dynamically-controlled scenario that is overwhelmingly dominated by reactions occurring at high temperatures or with unusual compositions. Even small contributions from processes occurring at lower temperatures or with more common compositions, which yield orders of magnitude higher equilibrium phosphate concentrations, would drive much higher bulk ocean phosphate abundance.

Both observational and modeling constraints suggest that high temperature (>200 °C) reactions that yield low equilibrium phosphate concentrations are unlikely to dominate ocean chemistry. Silica nanoparticles observed in the E-ring of Saturn are interpreted as evidence of 50-150 °C hydrothermal fluids[11], while groundwater simulations for tidally heated hydrothermal systems on Enceladus yield aquifer temperatures up to 90 °C, assuming an isotropic permeability of $10^{-14}$ m$^2$ and exceptionally high tidal dissipation within the core[34,59]. Similarly, groundwater simulations for radioactively heated hydrothermal systems on Europa yield aquifer temperatures around 100 °C[35]. Moreover, in our simulations, reactions occurring at temperatures >150 °C generally yield fluids that are more acidic than the alkaline conditions thought to prevail in the ocean of Enceladus (Supplemental Fig. 1), suggesting that high temperature fluids likely do not dominate the inputs to that ocean. For temperatures <150 °C, total dissolved phosphate exceeds $10^{-7}$ mol/kg for nominal CI chondrite composition for all modeled W:R (Fig. 2) and is consistently >$10^{-5}$ mol/kg for W:R < 100. For reference, the observed chemistry of high and low temperature hydrothermal vent fluids on Earth implies W:R of 0.1-10[45], reflecting moderate to extensive reaction and representing the low end of the much broader range that we have modeled here. Finally, high temperature (>200 °C) hydrothermal vents on Earth exist exclusively in areas of active volcanism, whereas the totality of crustal fluid flow is more widespread and occurs over a broader temperature range. Observational constraints on the global energy budget of ocean moons suggest that even if volcanism were to exist, it would likely be a relatively small portion of the global heat budget[37], and that extensive regions of lower temperature water-rock reaction could therefore be expected. Hence, the phosphate-limited high temperature region of our results is unlikely to dominate ocean chemistry.

We explored the implications of the modeled total dissolved phosphate concentrations for biology by reference to cellular phosphorus inventory and uptake kinetics in aquatic cell populations on Earth. It should be noted that these calculations are not predictions of cell abundance and doubling time for an inhabited Enceladus; rather, they simply convey the results of our abiotic equilibrium calculations in biologically tangible terms.

Upper limits on cell abundance were estimated by dividing model-predicted total dissolved phosphate by the mean per-cell phosphorus content of 336 representative cells sampled from several aquatic environments on Earth[54]. The computed values represent the volume-normalized number of Earth-like cells that could be synthesized if all aqueous phosphate was used for biosynthesis. Based on the mean value of 0.69 ± 0.38 fg P/cell[39], the modeled ranges of temperature and W:R ratios could support cell densities from $10^3$–$10^{11}$ cells/mL (Fig. 4a). For reference, the average cell density in Earth's deep oceans lies in the range of $10^5$–$10^6$ cells/mL[25], and modeled concentrations of total dissolved phosphate for nominal CI chondrite composition could support comparable or larger cell densities for all T < 200 °C and all W:R ratio <100. For comparison, estimates of energy-limited cell abundance on Enceladus are generally <$10^4$ cells/mL[24] although some recent studies suggest > $10^5$ cells/mL[60,61]. Calculated cell abundances of ~$10^2$ cells/mL at high temperatures, high W:R ratios, and for less common chondrite types are comparable to the densities observed in some of Earth's most oligotrophic settings, and to the benchmark value of cell densities considered by the Europa Lander science definition study[58]. Thus, total dissolved phosphate is unlikely to be limiting with respect to the establishment of detectable populations of Earth-like cells.

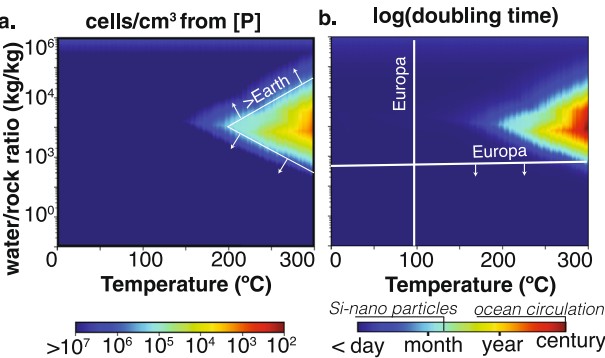

**Fig. 4 | Biological context of CI chondrite dissolved phosphate concentrations.** Cellular abundance **a** and doubling times **b** assuming Earth-like phosphorus concentrations within cells[54] and the most conservative *Prochlorococcus* uptake kinetics[19]. Relevant observational constraints for Enceladus and Europa are annotated[11,16,54,64].

It is also relevant to consider the rate at which concentrations of total dissolved phosphate, if limiting, would allow a population of cells to develop. The effect of dissolved phosphate concentrations on rates of biological phosphate uptake and cell growth has been extensively studied in phytoplankton populations across a range of sites in Earth's oceans. The resulting parameterizations of phosphate uptake rate as a function of concentration and cell biomass were used to calculate the P-limited rate of cell growth. We express the values as hypothetical "doubling times", which represent the time required for the cell abundance to increase two-fold. The calculated values correspond to initial growth rates that would decrease as the standing concentration of phosphate is drawn down by biosynthesis. They do not provide a basis for assessing steady-state productivity, which depends on fluxes that are not modeled here. Rather, they provide a reference point for comparing the hypothetical grow-in time of a population to the time scales associated with relevant physical and chemical processes, such as the ascension/dispersion of buoyant plume fluids through an ocean water column. We use the phosphate uptake parameterization given by Lomas and co-workers[19] for the photosynthetic picoplankton *Prochlorococcus* specifically to represent the sorts of uptake rates that are observed in phosphate-limited microbial populations. With this basis, we calculate hypothetical doubling times of days to weeks for nominal CI chondrite composition at most water-rock ratios and temperatures (Fig. 4b). For reference, the fastest doubling that can result from the parameterization used here is 1.5 days, and this rate is approached across a broad range of the temperatures and W:R we evaluated.

The foregoing discussion has sought to place the results of our abiotic equilibrium calculations in biologically tangible terms. However, should life exist on Enceladus, the biological activity could, itself, affect phosphate availability, with the potential to yield cell abundances that are either higher or lower relative to the values plotted in Fig. 4.

Higher cell abundances could result if active uptake of phosphate by biological activity serves to maintain its concentration in the oceans at levels below the equilibrium associated with geochemical processes. For reference, phytoplankton in Earth's oceans are capable of scavenging phosphate to sub-nanomolar levels orders of magnitude below the equilibrium phosphate concentrations associated with most of the modeled range of temperatures and W:R. In these circumstances, the dissolution of phosphate containing minerals would remain thermodynamically favorable and water-rock interactions would continue to serve as a source of phosphate to the ocean. Through time, this scenario could establish a standing biomass well in excess of the reference values in Fig. 4a. On Earth, coral reefs stand as an example of high-biomass systems that are built up through time from often nutrient-poor waters.

Lower cell abundances could result from cell attrition (e.g., via lysis or grazing) if the phosphate content of those "lost" cells is subsequently removed from the ocean system and becomes unavailable to fuel new cell growth. The "biological pump" in Earth's oceans is an example of such a removal mechanism. This process redistributes cellular organic matter, including organic phosphate, by packaging it into denser-than-water particles or pellets that sink or are transported from the surface to the deep ocean and underlying sediments. Microbial activity associated with sinking or sedimented particles serves to recycle most of the associated carbon, phosphorus, and other constituents to the dissolved phase, while a small fraction (<0.5% as an ocean-wide average)[62] becomes deeply buried and is effectively lost from the ocean system for long durations. The relative rates of long-term burial vs. dissolution of organic materials in the biological pump – its efficiency in removing phosphate from the system—is primarily a function of the sinking time for organic particles[62]. On ocean moons, reduced gravity and a deeper water column mean that particles with the same size and density would have a potentially much longer sinking time and therefore greater efficiency in returning phosphate to the dissolved phase. For example, the deeper water column and reduced gravity on Enceladus would result in ~1000-fold longer particle sinking times than for equivalent particles in Earth's oceans. Moreover, it is unclear whether biomineralization (e.g., formation of exoskeletons) or predation and the formation of fecal pellets—two of the most important mechanisms that form dense, rapidly sinking particles on Earth—should be expected on other ocean worlds. Finally, the absence of subaerial weathering on ocean moons implies the potential for greatly reduced sedimentation rates relative to those in Earth's oceans, with a corresponding potential for diminished burial efficiency. Considered collectively, these factors suggest that a biological pump-like mechanism might be a much less effective phosphate removal mechanism on ocean moons in comparison with its strong influence on the distribution of phosphate in Earth's oceans.

We have addressed the question of phosphate availability on ocean worlds by modeling the equilibrium phosphate concentrations associated with aqueous alteration of chondritic materials across a range of reaction conditions and compositional inputs. For the most common chondrite composition and the range of conditions that is most consistent with observational constraints at Enceladus and Europa, equilibrium phosphate concentrations equate to potential cell abundances greater than those in Earth's deep oceans and greater than estimates of energy-limited cell abundance on Enceladus. As such, we conclude that phosphate availability is likely not limiting to life on ocean worlds. Our findings complement the work of Hao and coworkers[20] by supporting a broadly similar conclusion concerning phosphate availability, using a distinct approach. By considering broad range of compositional inputs and reaction conditions, we show that, for the most plausible scenarios, this conclusion is robust even if bulk ocean phosphate is subject to significant dynamic control by reactions occurring at conditions much different than those in the bulk ocean.

## Methods
### Equilibrium geochemical models
Thermodynamic calculations were performed with the program EQ3/6, a thoroughly vetted and commonly used aqueous geochemistry code that computes the equilibrium distribution of dissolved and solid phase species based on a specified initial composition and reaction conditions. More specifically, the code uses the coupled equations of mass balance, charge balance, and mass action to compute an equilibrium distribution based on experimentally-constrained thermodynamic data for relevant aqueous and solid phase species. The thermodynamic data and parameters used in this study are drawn from work by Ely and references therein[45]. Our calculations simulated the reaction of prescribed mass ratios of "minimal fluid" with solid phase "special reactant," with the compositions of those two reacting phases formulated as follows:

For "minimal fluid," we used the EQ3 program to calculate the equilibrium concentrations and activities for all aqueous species in our database, at a specified pressure and temperature, in a solution consisting of neutral, anoxic water with a negligible elemental inventory of $10^{-13}$ mol/kg $H_2O$ for each of Al, C, Ca, Cl, F, Fe, K, Li, Mg, Mn, Na, P, S, and Si. ("Seeding" the calculation with a minimal initial inventory of elements increases numerical stability in the calculations.) For these calculations, the pressure was held constant at 50 MPa and a separate speciation calculation was performed for each of 300 different temperatures, ranging from 0 to 300 °C in one-degree increments. The outputs of these speciation calculations were then used as the aqueous reactant input for the water-rock reaction modeling. To parameterize the solid phase input, we used the "special reactant" function of EQ6, with one mole of special reactant comprising 1.00 kg of solid material with the elemental compositions shown in Supplemental Table 1. We calculated those compositions based on the element mass fractions of Al, C, Ca, Cl, Fe, K, Li, Mg, Mn, Na, P, S, and Si as reported in MetDB for each chondrite type. Cl is unreliably measured and so we use the concentration of 704 ppm from Lodders and co-workers[52] for all simulations. In each case, the oxygen content was determined by the molar quantity of $O^{2-}$ required to achieve charge balance in the special reactant given the molar abundance of each other constituent element, when those elements were assigned nominal oxidation states as follows: $Al^{3+}$, $C^{2+}$, $Ca^{2+}$, $Cl^-$, $F^-$, $Fe^{2+}$, $K^+$, $Li^+$, $Mg^{2+}$, $Mn^{2+}$, $Na^+$, $P^{5+}$, $S^{2-}$, and $Si^{4+}$. For the redox sensitivity analysis that was performed with CI chondrite composition, the oxygen mass fraction of reacting solids was set at 0.311, 0.351, and 0.388, while holding the mass fractions of all other solid phase inputs fixed at values equivalent to CI chondrite composition (Supplemental Table 1). This is equivalent to fixing the oxidation state of C at 0, 2 +, and 4+ while holding the oxidation states of S and Fe at 2- and 2 +, respectively, but it should be noted that the net effect of this parameterization is to vary the oxidation state of the solid phase material overall rather than that of carbon specifically. Once a fully charge-balanced special reactant was created, the relative element molar abundances were scaled in order to achieve a total mass of 1.00 kg.

The EQ6 program was used to simulate the reaction of each special reactant with minimal fluid at 50 MPa and temperatures ranging from 0 to 300 °C in one-degree increments, using the EQ3 output for equilibrated minimal fluid at the corresponding temperature (e.g., a water-rock reaction calculation performed at 50 °C used, as input, a minimal fluid that had been speciated at 50 °C in EQ3). In each simulation, the fluid and mineral composition was output at 1075 unique values of the reaction progress variable, $\xi_i$, which corresponds to the fraction of a specified mass of special reactant that has reacted with a specified mass of minimal fluid. Each value of xi thus corresponds to a unique W:R ratio, such that, for every combination of reactants and temperatures, the equilibrium distribution of aqueous and mineral species was determined for each of 1075 unique W:R ratios.

## Cell doubling times

For calculating hypothetical cell doubling times, we use the phosphate uptake kinetics parameterization of Lomas and co-workers[19]. Organisms on Earth vary widely in their phosphate uptake capabilities. The use of a phytoplankton parameterization is intended here to reflect what is possible in microbial populations that, by virtue of adaptation to phosphate limitation, are highly efficient in their use of phosphate. Among the several species and mixed populations of phytoplankton that were considered by Lomas and co-workers[19], we used the parameterization developed for the picoplankton *Prochlorococcus*, which yields lower rates than those observed in mixed species populations Lomas and co-workers[19] or predicted by previous work that used Michaelis-Menten type uptake kinetics[63]. In this respect, it yields phosphate uptake rates that are conservative with respect to the observed capabilities of natural populations on Earth. For detailed methodology please see Lomas and co-workers[19] methods.

## Data availability

These output files are provided in pickle format archived at (10.5281/zenodo.7683874) and readable using standard open-source data processing software (e.g., https://www.python.org/). The authors declare that the data supporting the findings of this study are available within the paper and supporting information files.

## Code availability

All simulations are produced using the open-source software EQ3/6[43,44] with ~300,000 model outputs for each of the 9 compositional inputs described in the text. We include a representative example input file that can be modified to reproduce all results archived at (10.5281/zenodo.7683874).

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

## Acknowledgements

Support for this study was provided by NASA Grant 80NSSC19K1427 (C.G., T.H., E.S.) and the NASA Planetary Science Division ISFM Program. N.R.F. and T.E. were supported through the NASA Postdoctoral Fellowship Program. We thank the Exploring Ocean Worlds team for feedback on improving the manuscript.

## Author contributions

N.R.F. and T.H. designed the project and wrote the manuscript in consultation with all authors. N.R.F. performed numerical models in consultation with T.E. and S.S. T.E. and E.S. developed thermodynamic database used for calculations. C.G. led comparison with Earth analogs. N.R.F., C.G., T.H., T.E., S.S., E.S. contributed to the interpretation of data and model results.

## Competing interests

The authors declare no competing interests.
