## [Peer Review File · Nature Communications]

Phosphate availability and implications for life on ocean worldsReviewer #1 (Remarks to the Author):

In this work, Randolph-Flagg et al. utilise a numerical model for estimating the equilibrium dissolved P concentration in putative ocean worlds as a function of certain input parameters (water-rock ratio; temperature). The central theme tackled is timely and important in astrobiology, it falls within the scope of the journal, and it has not been investigated before with the exception of Lingam & Loeb (2018). The analysis also appears to be robust, with some caveats that are discussed later. On the whole, I was glad to read this interesting paper.

A revised manuscript that addresses the following comments could be suitable for publication in this journal. As I will be touching on some aspects that overlap with my past work, I wish to disclose my identity at the outset - Manasvi Lingam

Major comments:

1. In the Introduction, there should be an explicit acknowledgment that one is interested in calculating phosphorus abundance in the form of phosphates specifically. It is actually (ortho)phosphate (rather than P in any generic form) that assumes vital roles in biology on Earth. Hence, the assessment of dissolved P should be interpreted accordingly.

2. The authors allude to biology (in connection with the P cycle) in lines 122-125. This is an important point that needs to be expanded on by adding least a few additional sentences.

The rationale is that sedimentation of particulate matter (including dead organisms) would constitute a major P sink; in fact, it is dominant on Earth, as seen from Fig. 2 in Paytan & McLaughlin (2007) [cited in the manuscript]. To put it differently, if organisms were to exist, they would use up some P and not all of this P would be remineralised after their demise, thence leading to a net loss of P. Hence, it is vital for the authors to convey that their estimates are potentially realistic in the absence of biology and that certain physical and chemical processes are not considered. Naturally, this limitation is applicable to other models of this ilk, as mentioned in Section 7.6.2 of Lingam & Loeb (2021):

<https://www.hup.harvard.edu/catalog.php?isbn=9780674987579>

3. What I deem the biggest drawback of the current version of the manuscript is fortunately (relatively) easy to fix. Apart from a brief allusion to "geochemical modeling software EQ3/6" (line 131), no other information regarding the code is provided. I was not provided access to any "Supplemental Information" (SI) where some details might have been included.

However, irrespective of whether the putative SI contains the salient materials, it is desirable for the main manuscript to sketch how the numerical model employed herein works. The reasons are twofold. First, even if the code is well-validated, the broad readership of this journal, who may not peruse the SI, would have no ideal of how it functions, thereby rendering it a black box. Second, because the audience is not readily acquainted with what input parameters have been entered and how, this issue may cause impediments while reproducing the results.

I recognise that furnishing an in-depth description of the code is manifestly impractical. If the authors can, however, create a table depicting the salient input parameters and a schematic figure of how the code operates, I believe that this should be sufficient information for the readers.

4. In line 197, the authors infer a water-rock ratio of <20 by combining an estimate from Postberg et al. (2009) with their simulation results for Na^+ . If one considers the bulk ocean - and not specialised environments such as hydrothermal vents that have low water-rock values - this ratio might be lowered. Could the authors comment on how robust their inference is?

Next, as a point of comparison, it would be helpful if they specify the average water-rock value for Earth's oceans (i.e., bulk value) somewhere in the paper. By adopting the

average water-rock ratio and temperature for Earth's oceans, it would be interesting (and worthwhile) to determine what equilibrium P concentration is obtained, and compare it with the bulk oceanic dissolved P concentration on Earth; the two would not necessarily match, but the comparison ought to be illuminating.

5. If ocean worlds (like Europa and Enceladus) were present at (or before) ~4 Ga, they could have received a high influx of metallic iron (oxidation state of zero) analogous to studies of early Earth (e.g., Zahnle et al. 2020); a fraction of this iron would be transported into the ocean. Hence, referring to line 206, how are the authors' results impacted if the redox state of iron is chosen to be "0"?

<https://iopscience.iop.org/article/10.3847/PSJ/ab7e2c/meta>

6. In line 223-224, the authors make a crucial assumption that all aqueous P is used for biosynthesis. By "P" here, do they mean phosphorus in any form or specifically as orthophosphate? [this question ties in with point #1 above] Moreover, the assumption of 100% utilization efficiency is rather extreme - plants like *Oryza sativa* have efficiencies closer to 10%. Hence, the authors should adopt values compatible with plankton and redo the analysis.

7. Lines 234-250: The "doubling times" thus calculated seem to take only P limitation into account, and not other factors (e.g., miscellaneous nutrients; energy). If so, this caveat should be noted in the paragraph. Furthermore, with regards to lines 245-246, is the chosen P-uptake parametrisation also dependent on the same input parameters (like temperature)? This aspect becomes crucial because P-uptake is anticipated to depend on temperature, pH, etc. In case the authors have used a temperature-independent P-uptake parametrisation, this limitation must also be specified.

Minor comments:

1. In line 41, the authors cite an upper bound of 11 for the pH of early Earth; I have not seen such an alkaline pH in the modern literature, e.g., the two modern references (2017/2018) cited by the authors have a pH of <9. It may be worth adjusting the limits or the phrasing accordingly.

2. Line 57: In tandem with Benner & Hutter (2002), the classic paper by Wertheimer (1987) on the significance of phosphates is worth citing here:
<https://www.science.org/doi/abs/10.1126/science.2434996>

3. Line 74: Typo in "Lingham", which must be changed to "Lingam"

4. The discussion in lines 109-122 conveys the basic point nicely. In a "zeroth" order model, one may assume that all the P is depleted near a hydrothermal vent but, as line 116 indicates, about 10% of the original concentration survives.

5. In connection with Fig. 3 and the accompanying discussion, the authors should compare this estimate (derived from P limitation) with others based on energetic considerations. A number of such studies exist by Hand, Vance, Steel, McKay, and others - which are cited in Chapter 7.5 of Lingam & Loeb (2021) - and they usually yield cell densities of > 1000 cells/mL.

Reviewer #2 (Remarks to the Author):

The manuscript addresses the availability of phosphorous (P) from water-rock reactions in ocean worlds. By considering the likely bulk compositions of ocean worlds and the associated water-rock reactions, the authors arrive at abundances comparable to or exceeding those in Earth's deep oceans. By analogy to life on Earth, this might imply that cell numbers in ocean worlds could exceed those in deep ocean environments on

Earth.

I find the manuscript to be informative, if parochial to problems related to P. I think the work should be published with the current focus on P, but only after addressing some inconsistencies and editorial issues.

The paper responds in a multiple passages to predictions by Lingham and Loeb (2019), a paper that provides a generalized overview of problems pertaining to planetary habitability. The authors should elaborate on the detailed logic behind the conclusions offered by Lingham and Loeb, and should ground those conclusions in findings from studies in Earth systems or in the laboratory. In addition, including discussion of planetary geochemical models from Neveu et al. (2017) and Melwani Daswani et al. (2021) would provide important context for the assumed alteration pathways to generate P. Notably, the latter reference also discusses in detail the likely chondritic composition of Europa.

-Steven Vance

other comments:

Line 30: change "could potentially" to "could" or "might"

Line 33: the last sentence of this paragraph is overly broad. Evidence for an ocean at Triton is scant and circumstantial based on assumed tidal heating. Nearly all available information about Triton comes from the Voyager spacecraft.

Line 87: change "are" to "is"

Line 116: I find the use of the term "sinks" confusing, unless 5×10^{-6} should be 5×10^{-8} . My confusion also stems from the earlier caveat that "sink" is a relative term.

Line 144: please explain the effects of chemistry at higher pressures—relevant minimum pressures at Europa may exceed 150 MPa, and pressures in the deep interior could approach 1 GPa. In larger worlds such as Titan, 1 GPa could be the minimum pressure for water-rock chemistry.

Line 193: M is used to denote phosphorous concentration, whereas everywhere else in the paper mol/kg is used.

Line 201: also cite Hand and Chyba 2007

Line 216: remove ", if it exists".

Reviewer #3 (Remarks to the Author):

Dear Authors and Editor,

Please find below my review for the manuscript Phosphorus availability and implications for life on ocean worlds, by Randolph-Flagg and coauthors.

The paper describes an investigation to predict the presence and concentration of phosphorus in ocean worlds, to test whether sufficient phosphorus would be present to sustain cellular life as we know it. Of note, phosphorus has not been definitively

detected at ocean worlds (which is why such predictive modeling studies are important). The study used thermodynamic equilibrium models to compute the equilibrium composition of pure water reacting with six carbonaceous chondrite compositions, which could plausibly be representative of rocks on ocean worlds. The results show that for all chondritic compositions, the resulting phosphorus concentrations under the conditions likely to exist at Enceladus' ocean exceed the concentration requirement for certain oceanic microorganisms. Therefore, phosphorus would not be a limiting factor to the habitability of Enceladus.

It is my opinion that the paper is overall well-written and clear. The methodology is well established and sound, and the results mostly support the discussion and conclusions presented. However, it is my subjective opinion that the work presented is not especially significant, or noteworthy enough to be published in Nature Communications. My main concern is that similar work has been carried out for decades, and this work appears to be a new iteration on an old theme. The paper appears sound, but perhaps, the preferred journals to target could be *Icarus*, *Meteoritics and Planetary Science*, *Geochimica et Cosmochimica Acta*, *JGR:Planets*, *Astrobiology*, or a similar, specialized journal. The paper does not clarify how the methods and models carried out are significantly different from previous efforts, particularly Zolotov et al. (2007) "An oceanic composition on early and today's Enceladus", *Geophys. Res. Lett.* 34, L23203, doi:10.1029/2007GL031234 (but also other papers). This new paper explores more compositions, and then uses the results for astrobiological implications, however, the methods otherwise seem rather similar, as far as I understand.

Another problem I notice is that the paper broadly suggests that the conditions tested in the models apply to many ocean worlds, when it mostly just applies to Enceladus. I appreciate that, importantly, various redox states for the iron and carbon were tested in the models, but unfortunately, the redox state of the resulting dissolved phosphorus is not reported in the paper (even if the total concentration of dissolved phosphorus remained roughly constant). For example, phosphorus in Titan has been predicted to be reduced (PH₃), and therefore, could be effectively trapped in clathrate hydrates (see a paper that was very unfortunately omitted from this manuscript: Pasek et al., 2011; "Phosphorus chemistry on Titan" *Icarus* 212, 751-761; 10.1016/j.icarus.2011.01.026. Further quantitative constraints about phosphorus in ocean worlds are considered there).

In addition, because the paper deals with equilibrium models, then it is of interest to understand the full phase and species assemblage, including pH and redox potential. However, the paper only presents results about phosphorus concentration. It would be beneficial to also see the resulting mineral concentrations and other species in solution, as a function of the variables tested (namely water-to-rock ratio and temperature). Unfortunately, it is not possible to evaluate the quality of the models without presenting this information (it should at the very least be in supporting information). For example, it would have been very useful to see the tradeoff between gypsum (CaSO₄), calcite and apatite, since they would all compete for calcium. It is also typical to explain whether any mineral species or chemical reactions were suppressed because they are kinetically inhibited under the conditions modeled (for example, if methane forms from CO₂-H₂-H₂O equilibrium, then carbonate may be removed from solution, so calcium can sequester phosphate instead of carbonate). The paper also did not sufficiently describe what the compositions of the carbonaceous chondrite reactant rocks were. A table with these compositions would have been helpful. I think that a little more space (afforded by other journals) to incorporate this important information would have benefited this paper.

Finally, the paper did not compare the results of the models (particularly Figure 2) to the pH constraints from Enceladus. Do all the water-to-rock ratios and temperatures tested and shown in the figure yield pH consistent with that inferred from Cassini data?

Some other comments:

Lines 47-52 omit the fact that the “biogenic element” sulfur has also not been definitively detected at Enceladus either. As such, the case for focusing solely on phosphorus is weakened.

Figure 1: The significance of this figure is not sufficiently or adequately described in the paper. I do not understand why presenting information about these particular vents is meaningful. Why these and not others? Are hydrothermal vents on Earth even good analogs for vents on ocean worlds? Are they better analogs to vents at Enceladus than the hydrothermal alteration experiments of chondrites (e.g. Kikuchi et al. 2022; *Geochimica et Cosmochimica Acta* 319, 151–167; Suttle et al. 2022 *Geochimica et Cosmochimica Acta* 318, 83–111)? What does NESCA mean? It is not spelled out in the paper. I think a table presenting the mean composition (and standard deviation) of the compositions of the chondrites and MORB would have been more quantitative and useful than Figure 1a, especially since it is not possible to distinguish between the chondrites in the figure.

Figure 2: This is a genuinely interesting figure, but I do not see constraints for pH. The models should have yielded pH values. Some regions would likely yield pH values inconsistent with those inferred at Enceladus from Cassini data.

REVIEWER COMMENTS

Reviewer #1 (Remarks to the Author):

In this work, Randolph-Flagg et al. utilise a numerical model for estimating the equilibrium dissolved P concentration in putative ocean worlds as a function of certain input parameters (water-rock ratio; temperature). The central theme tackled is timely and important in astrobiology, it falls within the scope of the journal, and it has not been investigated before with the exception of Lingam & Loeb (2018). The analysis also appears to be robust, with some caveats that are discussed later. On the whole, I was glad to read this interesting paper.

A revised manuscript that addresses the following comments could be suitable for publication in this journal. As I will be touching on some aspects that overlap with my past work, I wish to disclose my identity at the outset - Manasvi Lingam

Thank you so much for the interest in our work and for your own foundational work on this topic.

Major comments:

1. In the Introduction, there should be an explicit acknowledgment that one is interested in calculating phosphorus abundance in the form of phosphates specifically. It is actually (ortho)phosphate (rather than P in any generic form) that assumes vital roles in biology on Earth. Hence, the assessment of dissolved P should be interpreted accordingly.

The revised text addresses this distinction in the introduction, results, and throughout the text. Specifically, the introduction refers to both 'phosphorus' and 'phosphate' in order to survey a relevant literature that, in some cases, refers to phosphate specifically and, in others, refers to phosphorus in other oxidation states. In the results, we note that our model includes the possibility for P speciation into other oxidation states but that orthophosphates are shown to be the dominant form (>99%) at equilibrium over the full range of modeled conditions. All of the subsequent text refers specifically to total dissolved phosphate.

2. The authors allude to biology (in connection with the P cycle) in lines 122-125. This is an important point that needs to be expanded on by adding at least a few additional sentences.

The rationale is that sedimentation of particulate matter (including dead organisms) would constitute a major P sink; in fact, it is dominant on Earth, as seen from Fig. 2 in Paytan & McLaughlin (2007) [cited in the manuscript]. To put it differently, if organisms were to exist, they would use up some P and not all of this P would be remineralised after their demise, thence leading to a net loss of P. Hence, it is vital for the authors to convey that their estimates are potentially realistic in the absence of biology and that certain physical and chemical processes are not considered. Naturally, this limitation is

applicable to other models of this ilk, as mentioned in Section 7.6.2 of Lingam & Loeb (2021):

<https://www.hup.harvard.edu/catalog.php?isbn=9780674987579>

This is a very important point and, indeed, a key intent of our work is to emphasize that both equilibrium chemistry and dynamical factors, including biological processes, can exert controls on dissolved phosphate. An advantage of the transfer of this manuscript to Nature Communications is that we now have the space to expand our discussion in this area and we appreciate the suggestion to do so. We have added a new section to the discussion that specifically addresses the role of biological processes as follows:

(i) we note that the calculated cell abundances and hypothetical doubling times should be viewed not as predictions for an inhabited Enceladus but, rather, as points of reference that can convey the results of *abiotic* equilibrium calculations in biologically-tangible terms.

(ii) we describe how biological activity (including a mechanism like Earth's biological pump) could potentially yield cell abundances both higher and lower than the abiotic reference points.

(iii) we briefly describe the factors that influence the efficiency with which Earth's biological pump removes phosphate from the ocean and consider how such efficiency might differ on a world like Enceladus.

3. What I deem the biggest drawback of the current version of the manuscript is fortunately (relatively) easy to fix. Apart from a brief allusion to "geochemical modeling software EQ3/6" (line 131), no other information regarding the code is provided. I was not provided access to any "Supplemental Information" (SI) where some details might have been included.

However, irrespective of whether the putative SI contains the salient materials, it is desirable for the main manuscript to sketch how the numerical model employed herein works. The reasons are twofold. First, even if the code is well-validated, the broad readership of this journal, who may not peruse the SI, would have no ideal of how it functions, thereby rendering it a black box. Second, because the audience is not readily acquainted with what input parameters have been entered and how, this issue may cause impediments while reproducing the results.

I recognise that furnishing an in-depth description of the code is manifestly impractical. If the authors can, however, create a table depicting the salient input parameters and a schematic figure of how the code operates, I believe that this should be sufficient information for the readers.

The revised text now includes both a brief description of the modeling software in the body of the results and a dedicated Methods section that encompasses an expanded description of that software and the details of its specific application in this study. A supplemental material section has been added to detail the model input parameters (compositional inputs and thermodynamic data).

4. In line 197, the authors infer a water-rock ratio of <20 by combining an estimate from Postberg et al. (2009) with their simulation results for Na+. If one considers the bulk ocean - and not specialised environments such as hydrothermal vents that have low water-rock values - this ratio might be lowered. Could the authors comment on how robust their inference is?

Postberg et al (2009 and 2011) interpret the observation of Na in plume materials as originating from a liquid reservoir that presents a large (> sq km) evaporating surface at the base of the ice shell and subsequent works (e.g., Cable et al., 2020) have seen this as representing the bulk ocean. Accordingly, our usage applies the Na constraint in reference to the bulk ocean, rather than to, e.g., hydrothermal vent fluids, and we have expanded the revised text to clarify this. Independent work (e.g., Glein & Waite, 2020) places the bulk ocean water:rock ratio in the range 0.3-1.0, and the revised text introduces this estimate as an additional point of reference.

Next, as a point of comparison, it would be helpful if they specify the average water-rock value for Earth's oceans (i.e., bulk value) somewhere in the paper. By adopting the average water-rock ratio and temperature for Earth's oceans, it would be interesting (and worthwhile) to determine what equilibrium P concentration is obtained, and compare it with the bulk oceanic dissolved P concentration on Earth; the two would not necessarily match, but the comparison ought to be illuminating.

This is an interesting idea but complicated further by the fact that Earth has subaerial continents as well as oceans. An important first consequence is that the presence of large on-land NaCl evaporites and other solid phase sodium salt deposits precludes the use of modern-day bulk ocean Na contents as a conservative tracer of water:rock ratio for Earth's oceans in the same way that we use it, here, for Enceladus. Secondly, as the reviewer notes above, biology exerts a strong dynamical control on the distribution of phosphate, such that equilibrium with respect to hydrothermal alteration of basalt is not expected for Earth's oceans. Accordingly, we feel that it might needlessly complicate the discussion to present this comparison for Earth's oceans only to immediately work through a careful explanation of why agreement between equilibrium and observed phosphate concentrations is not to be expected anyway. For these reasons, we have chosen not to pursue this suggested comparison to Na and P concentrations in Earth's oceans.

5. If ocean worlds (like Europa and Enceladus) were present at (or before) ~4 Ga, they could have received a high influx of metallic iron (oxidation state of zero) analogous to studies of early Earth (e.g., Zahnle et al. 2020); a fraction of this iron would be transported into the ocean. Hence, referring to line 206, how are the authors' results impacted if the redox state of iron is chosen to be "0"?

<https://iopscience.iop.org/article/10.3847/PSJ/ab7e2c/meta>

This is a very interesting question and we have taken it as motivation to expand our redox sensitivity analysis to significantly more reducing conditions than we previously considered. As now described in the new methods section, the parameterization of redox state in reacted

materials is achieved by varying the oxygen mass fraction relative to fixed mass fractions of other elements in the reacted materials. In the sensitivity analysis, we vary the oxygen mass fraction from 0.311 to 0.388, which is equivalent to varying the redox state of all carbon in the system from zero to 4+, with all iron fixed at 2+ and all sulfur fixed at 2-. Relative to a nominal parameterization with carbon at 2+, the lowest oxygen mass fraction is approximately equivalent to ascribing an oxidation state of zero to all iron in the system (which would give an oxygen mass fraction of 0.305). We believe this range captures well the potential for conditions that are significantly more reducing in both bulk terms and in a localized context where a ‘late veneer’ effect might result in conditions even more reducing than the bulk average. To place this range in context, Zahnle et al estimated that the late veneer flux to Earth represents a quantity of iron equivalent to 0.16% of the planet’s mass. If delivered to a planet with an initial bulk iron content of about 7% (equivalent to chondrite composition), this would represent an addition of 2.3% to the total iron content. If all of the late veneer iron came in the metallic form, this would imply a change in the mean iron redox state of -0.05 (e.g., +2 to +1.95). Thus, we believe that a late veneer-like effect is well captured by the range now considered in our redox sensitivity analysis. That analysis does show diminished phosphate concentrations at intermediate W:R ratios for the most reducing condition, so we appreciate the suggestion to extend the analysis in that direction.

6. In line 223-224, the authors make a crucial assumption that all aqueous P is used for biosynthesis. By "P" here, do they mean phosphorus in any form or specifically as orthophosphate? [this question ties in with point #1 above] Moreover, the assumption of 100% utilization efficiency is rather extreme - plants like *Oryza sativa* have efficiencies closer to 10%. Hence, the authors should adopt values compatible with plankton and redo the analysis.

In the revised text, we characterize both the cell abundance and doubling time calculations as “potential” (e.g., “potential cell abundance”) and, as noted in the response to Comment #2, point out that the calculations are not intended as predictions but as points of reference meant to convey the results of abiotic equilibrium calculations in biologically-tangible terms. As such, we chose a basis for the calculations that would be concrete and easily grasped, and have attempted to convey in the revised text that the “potential” values we present should be seen in this light.

Regarding the specific form of P and the efficiency of phosphate utilization: In the revised version, the phrase “all aqueous P” is replaced by “total dissolved phosphate” in order to be specific to the form in question. Phosphate uptake capacities certainly can vary. For a reference calculation such as the one we offer, it was necessary to choose a specific value. The choice of 100% uptake was based on the observation that, for aquatic cell populations (phytoplankton), half-saturation constants for phosphate uptake are in the low nanomolar range (Lomas et al., 2014) – meaning that uptake occurs at fully half of its maximum rate at phosphate concentrations that are 2+ orders of magnitude lower than the equilibrium values that characterize most of our surveyed range. The implication is that such cells would continue to take up phosphate actively after having incorporated 99+% of the available pool.

7. Lines 234-250: The "doubling times" thus calculated seem to take only P limitation into account, and not other factors (e.g., miscellaneous nutrients; energy). If so, this caveat should be noted in the paragraph. Furthermore, with regards to lines 245-246, is the chosen P-uptake parametrisation also dependent on the same input parameters (like temperature)? This aspect becomes crucial because P-uptake is anticipated to depend on temperature, pH, etc. In case the authors have used a temperature-independent P-uptake parametrisation, this limitation must also be specified.

The revised text now includes a more detailed description of the Lomas model (the basis for our potential doubling time calculations) in the methods section and specifies, in the discussion section, that the calculations result from a parameterization that is specific to phosphate-limited phytoplankton populations. That parameterization does not include a temperature dependence but does result from, and is validated relative to, observations made across a range of environmental temperatures.

Minor comments:

1. In line 41, the authors cite an upper bound of 11 for the pH of early Earth; I have not seen such an alkaline pH in the modern literature, e.g., the two modern references (2017/2018) cited by the authors have a pH of <9. It may be worth adjusting the limits or the phrasing accordingly.

The introductory text has been changed so that it refers specifically to the more moderate pH and the two modern references. We also added Fig. S1 to compare pH values from simulations to the constraints and Fig. 3 to consider the range of redox conditions possible on these worlds.

2. Line 57: In tandem with Benner & Hutter (2002), the classic paper by Wertheimer (1987) on the significance of phosphates is worth citing here:

<https://www.science.org/doi/abs/10.1126/science.2434996>

Added, thank you!

3. Line 74: Typo in "Lingham", which must be changed to "Lingam"

Changed. Sorry!

4. The discussion in lines 109-122 conveys the basic point nicely. In a "zeroth" order model, one may assume that all the P is depleted near a hydrothermal vent but, as line 116 indicates, about 10% of the original concentration survives.

Thank you, we're glad this point came through!

5. In connection with Fig. 3 and the accompanying discussion, the authors should compare this estimate (derived from P limitation) with others based on energetic considerations. A number of such studies exist by Hand, Vance, Steel, McKay, and others - which are cited in Chapter 7.5 of Lingam & Loeb (2021) - and they usually yield cell densities of > 1000 cells/mL.

Indeed, a range of such estimates exist and, in almost all cases, the predicted cell abundances are well below the phosphate-limited “potential cell abundances” computed for most of the range of conditions that we modeled. One conclusion of our work is that other factors, such as energy availability, may prove more limiting to cell abundance on icy moons than nutrient availability. We have tried to make this point more clearly in the revised text by specifically pointing to the works on energy-limited cell abundance (as summarized by Cable et al., 2020) and noting that they generally predict abundances lower than those in our study.

Reviewer #2 (Remarks to the Author):

The manuscript addresses the availability of phosphorous (P) from water-rock reactions in ocean worlds. By considering the likely bulk compositions of ocean worlds and the associated water-rock reactions, the authors arrive at abundances comparable to or exceeding those in Earth's deep oceans. By analogy to life on Earth, this might imply that cell numbers in ocean worlds could exceed those in deep ocean environments on Earth.

I find the manuscript to be informative, if parochial to problems related to P. I think the work should be published with the current focus on P, but only after addressing some inconsistencies and editorial issues.

The paper responds in a multiple passages to predictions by Lingam and Loeb (2019), a paper that provides a generalized overview of problems pertaining to planetary habitability. The authors should elaborate on the detailed logic behind the conclusions offered by Lingam and Loeb, and should ground those conclusions in findings from studies in Earth systems or in the laboratory. In addition, including discussion of planetary geochemical models from Neveu et al. (2017) and Melwani Daswani et al. (2021) would provide important context for the assumed alteration pathways to generate P. Notably, the latter reference also discusses in detail the likely chondritic composition of Europa.

Thank you for your interest in our work and for the constructive feedback.

A key contribution of Lingam and Loeb (2018) to the topic of ocean world phosphate availability was to emphasize the importance of dynamical controls on oceanic P and highlight the important differences in how such controls might operate on icy moons, in a model informed by reference to Earth's phosphorus cycle. In the revised text we have attempted to elaborate on this work in two ways: (i) a stronger focus on the importance of integrating geochemical

equilibrium constraints with dynamical constraints, with discussion of the latter informed by reference to Lingam and Loeb (2018). The emphasis on dynamical controls is built into the revised text of both the introduction and the discussion. (ii) the addition of a section on biological processes as a dynamical control on phosphate, which again references the logic/framework in Lingam and Loeb (2018).

-Steven Vance

other comments:

Line 30: change "could potentially" to "could" or "might"

Changed

Line 33: the last sentence of this paragraph is overly broad. Evidence for an ocean at Triton is scant and circumstantial based on assumed tidal heating. Nearly all available information about Triton comes from the Voyager spacecraft.

We agree with this point and have changed the wording to reflect the range of proposed ocean worlds in Hendrix et al. (2018) rather than advocating for a specific ocean world.

Line 87: change "are" to "is"

Changed

Line 116: I find the use of the term "sinks" confusing, unless 5×10^{-6} should be 5×10^{-8} . My confusion also stems from the earlier caveat that "sink" is a relative term.

Good point. This language was significantly rewritten for clarity highlighting that lowering water column concentrations within the region of the plume by several hundred nanomoles/kg is relative to deep ocean concentrations and mostly a factor of .5 (Feely et al., 1996).

Line 144: please explain the effects of chemistry at higher pressures—relevant minimum pressures at Europa may exceed 150 MPa, and pressures in the deep interior could approach 1 GPa. In larger worlds such as Titan, 1 GPa could be the minimum pressure for water-rock chemistry.

The revised text includes a brief discussion of the extent to which our results are or are not applicable to various ocean worlds, with an acknowledgement that our models are run at a pressure (50 MPa) that is most relevant to aquifer conditions proposed for Enceladus in Chobet et al., 2018 and . Therein, we specify that the much different conditions (including pressure) on large moons like Ganymede and Titan place them beyond the scope of the present study.

The most robust thermodynamic data used in these simulations are for Earth abyssal depths (40-60 MPa) and we feel confident about the experimental data underlying these more moderate pressures and feel that the results are robust up to ~100 MPa. More sparse trench-depth data exist at pressures of 400+ MPa (e.g., Zhu and Sverjensky, 1991). There are theoretical ways to extend these constraints to higher and lower pressures which suggest that results are unlikely to vary by more than a factor of 2 or so (e.g., Hao et al., 2022). However,

given the feedback among different minerals we feel more confident focusing on the most relevant and best constrained pressure conditions.

Line 193: M is used to denote phosphorous concentration, whereas everywhere else in the paper mol/kg is used.

We have changed to units of mol/kg throughout.

Line 201: also cite Hand and Chyba 2007

Cited.

Line 216: remove ", if it exists".

Removed

Reviewer #3 (Remarks to the Author):

Dear Authors and Editor,

Please find below my review for the manuscript Phosphorus availability and implications for life on ocean worlds, by Randolph-Flagg and coauthors.

The paper describes an investigation to predict the presence and concentration of phosphorus in ocean worlds, to test whether sufficient phosphorus would be present to sustain cellular life as we know it. Of note, phosphorus has not been definitively detected at ocean worlds (which is why such predictive modeling studies are important). The study used thermodynamic equilibrium models to compute the equilibrium composition of pure water reacting with six carbonaceous chondrite compositions, which could plausibly be representative of rocks on ocean worlds. The results show that for all chondritic compositions, the resulting phosphorus concentrations under the conditions likely to exist at Enceladus' ocean exceed the concentration requirement for certain oceanic microorganisms. Therefore, phosphorus would not be a limiting factor to the habitability of Enceladus.

Thank you for your interest and constructive review.

It is my opinion that the paper is overall well-written and clear. The methodology is well established and sound, and the results mostly support the discussion and conclusions presented. However, it is my subjective opinion that the work presented is not especially significant, or noteworthy enough to be published in Nature Communications. My main concern is that similar work has been carried out for decades, and this work appears to

be a new iteration on an old theme. The paper appears sound, but perhaps, the preferred journals to target could be *Icarus*, *Meteoritics and Planetary Science*, *Geochimica et Cosmochimica Acta*, *JGR:Planets*, *Astrobiology*, or a similar, specialized journal. The paper does not clarify how the methods and models carried out are significantly different from previous efforts, particularly Zolotov et al. (2007) "An oceanic composition on early and today's Enceladus", *Geophys. Res. Lett.* 34, L23203, doi:10.1029/2007GL031234) (but also other papers). This new paper explores more compositions, and then uses the results for astrobiological implications, however, the methods otherwise seem rather similar, as far as I understand.

With the increased length of *Nature Communications*, we have attempted to expand on what we perceive to be the timeliness, importance, and novelty of our study. In particular (as noted in the revised introduction), the recent decadal survey in planetary science and astrobiology prioritized missions (flagship and *New Frontiers*) that would seek evidence of life on Enceladus, and this creates an impetus to consider the habitability of that world from all angles.

Previous work has, indeed, used similar methodology (including EQ3/6) to address questions of ocean chemical evolution on icy moons and it was purposeful to stick with what has been a workhorse approach. The perceived novelty that we have tried to convey in the revised text is: (i) A specific focus on phosphate chemistry that includes careful attention to the parameterization of phosphate chemistry in the thermodynamic database that supported this work. Zolotov (2007) predicted very low phosphate abundance when modeling the evolution of the Enceladus ocean. Here, our parameterization yields significantly higher phosphate abundance, similar to the findings of Hao et al (2022), which came out while we were revising our manuscript. (ii) Consideration of a broader range of compositional inputs and reaction conditions in order to evaluate the possibility that dynamical controls could drive bulk ocean phosphate concentrations to depart significantly from equilibrium with respect to the conditions therein. (iii) An effort to contextualize the results in biological terms to a greater extent than in previous work. In the expanded discussion, this includes a section on how biology, itself, could exert dynamical control.

Another problem I notice is that the paper broadly suggests that the conditions tested in the models apply to many ocean worlds, when it mostly just applies to Enceladus. I appreciate that, importantly, various redox states for the iron and carbon were tested in the models, but unfortunately, the redox state of the resulting dissolved phosphorus is not reported in the paper (even if the total concentration of dissolved phosphorus remained roughly constant). For example, phosphorus in Titan has been predicted to be reduced (PH₃), and therefore, could be effectively trapped in clathrate hydrates (see a paper that was very unfortunately omitted from this manuscript: Pasek et al., 2011; "Phosphorus chemistry on Titan" *Icarus* 212, 751-761; 10.1016/j.icarus.2011.01.026. Further quantitative constraints about phosphorus in ocean worlds are considered there).

Part of our rationale for considering a broad range of compositional inputs and reaction conditions was to produce results that are applicable across multiple ocean worlds but we agree that the results are most directly applicable to Enceladus. In our revised text, we note that this is a primary focus in part due to the emphasis placed on Enceladus by the recent NASEM decadal survey and in part because the observational constraints on Enceladus chemistry provide a stronger basis for grounding our results. We also include a paragraph in the discussion to describe the extent to which the conditions we modeled may be applicable to worlds other than Enceladus. In that paragraph, we note that application to Ganymede and Titan lie beyond the scope of the present study due to the very different conditions in pressure and hydrocarbon chemistry that prevail there.

In the results section, we now specify that the modeling includes the potential for P speciation into different redox states but that orthophosphate dominates (>99%) across the full range of modeled conditions. We note in the revised text that Hao et al (2022) observed a similar dominance of orthophosphate in their calculations.

In addition, because the paper deals with equilibrium models, then it is of interest to understand the full phase and species assemblage, including pH and redox potential. However, the paper only presents results about phosphorus concentration. It would be beneficial to also see the resulting mineral concentrations and other species in solution, as a function of the variables tested (namely water-to-rock ratio and temperature). Unfortunately, it is not possible to evaluate the quality of the models without presenting this information (it should at the very least be in supporting information). For example, it would have been very useful to see the tradeoff between gypsum (CaSO₄), calcite and apatite, since they would all compete for calcium. It is also typical to explain whether any mineral species or chemical reactions were suppressed because they are kinetically inhibited under the conditions modeled (for example, if methane forms from CO₂-H₂-H₂O equilibrium, then carbonate may be removed from solution, so calcium can sequester phosphate instead of carbonate). The paper also did not sufficiently describe what the compositions of the carbonaceous chondrite reactant rocks were. A table with these compositions would have been helpful. I think that a little more space (afforded by other journals) to incorporate this important information would have benefited this paper.

In the expanded results and discussion and new supplement we show the mineral species (Fig. 2), pH and redox conditions (Fig. 3, Supplemental Fig. 1), as well as Ca and Fe concentrations (Supplemental Fig. 1). No reactions are chemically inhibited as we view future experimental and observational data necessary to constrain these values. We hope that this study will help motivate more thorough databases of particularly S- and P- bearing mineral constants at a range of temperatures. In the expanded discussion we also discuss the role of gypsum in moderating Ca and P concentrations in these oceans. Finally we include the relevant thermodynamic model inputs in the supplement and accompanying data files.

Finally, the paper did not compare the results of the models (particularly Figure 2) to the pH constraints from Enceladus. Do all the water-to-rock ratios and temperatures tested and shown in the figure yield pH consistent with that inferred from Cassini data?

Supplemental Figure S1 was added to show the pH that results from reacting “nominal” CI chondrite composition across a range of T and W:R at a range of redox conditions (Fig. 3). As suggested by the reviewer’s comment, some parts of the modeled range are consistent with Enceladus ocean pH. Specifically, it is the lower range of reaction temperatures that yield alkaline pH values consistent with those inferred for the Enceladus ocean. The revised discussion makes note of this in a section that considers our model results in relation to observational constraints. Specifically, we note that bulk ocean pH constraints do not rule out contributions from reactions that yield more acidic fluids but they do suggest that such reactions are likely not the exclusive or dominant contributors to ocean chemical evolution.

Some other comments:

Lines 47-52 omit the fact that the “biogenic element” sulfur has also not been definitively detected at Enceladus either. As such, the case for focusing solely on phosphorus is weakened.

The decision to leave S out of the sentence in question was intended to keep the discussion streamlined and specific to the question of P availability, a focus that is motivated by several recent works on that topic. We felt that a discussion of S availability that included both Enceladus and Europa was not necessary in establishing the motivation for the present work and would distract from its primary focus.

Figure 1: The significance of this figure is not sufficiently or adequately described in the paper. I do not understand why presenting information about these particular vents is meaningful. Why these and not others? Are hydrothermal vents on Earth even good analogs for vents on ocean worlds? Are they better analogs to vents at Enceladus than the hydrothermal alteration experiments of chondrites (e.g. Kikuchi et al. 2022; *Geochimica et Cosmochimica Acta* 319, 151–167; Suttle et al. 2022 *Geochimica et Cosmochimica Acta* 318, 83–111)? What does NESCA mean? It is not spelled out in the paper. I think a table presenting the mean composition (and standard deviation) of the compositions of the chondrites and MORB would have been more quantitative and useful than Figure 1a, especially since it is not possible to distinguish between the chondrites in the figure.

Figure 1 is now encompassed in a slightly expanded introduction, with the purpose of illustrating (1a) that bulk P abundance is relatively similar across both a range of chondrites and MORB and (1b) that, as a tangible empirical point of reference, hydrothermal vents on Earth do not remove phosphate to zero but, rather, to a level that is substantial relative to the requirements and uptake potential of aquatic microorganisms in Earth’s oceans. Regarding the data in Fig. 1b, phosphate has not frequently been measured in vent fluids. To our knowledge, Fig 1b is complete with respect to existing measurements of vent fluid phosphate. The caption for Fig 1

now specifies that NESCA refers to the (MORB-hosted) Northern Escanaba Trough site on the Gorda Ridge.

Thank you for the suggestion of recent references relating to the chondrite alteration experiments, which have now been included in the revised text.

Figure 2: This is a genuinely interesting figure, but I do not see constraints for pH. The models should have yielded pH values. Some regions would likely yield pH values inconsistent with those inferred at Enceladus from Cassini data.

Figure S1 addresses these pH observational constraints at temperature. It should be noted that cooling and mixing of hydrothermal fluids in the ocean may change the bulk ocean phosphate.

Reviewer #1 (Remarks to the Author):

The authors have done a meticulous and satisfactory job of addressing my comments, as well as those of the other reviewers, and I commend them for doing so. I would like the authors to tackle the following two minor matters, after which the manuscript ought to be suitable for publication.

1. As the authors would have noticed, the paper by Hao et al. (2022) appeared in PNAS subsequent to their initial submission, which contains similar (but not identical) results. Therefore, I would like to see a moderately expanded discussion (in the Introduction) of how the framework employed by the authors diverges from and/or overlaps with Hao et al. (2022). This discussion is important in order to highlight the novel aspects of the authors' approach, as well as to underscore how the two studies broadly agree with each other.

2. With regard to lines 324-326, there are certain studies that have obtained cell densities greater than 10,000 cells/mL. One recent example is Affholder et al. (2022) - which incorporates both nutrient and energetic constraints - and some additional studies and simple calculations are provided in Chapter 7.5 of Lingam & Loeb (2021).

<https://iopscience.iop.org/article/10.3847/PSJ/aca275/meta>

<https://www.hup.harvard.edu/catalog.php?isbn=9780674987579>

Hence, I would recommend modifying lines 324-326 slightly, and citing the aforementioned reference(s).

Reviewer #2 (Remarks to the Author):

The authors have addressed all of my concerns. The manuscript is ready for publication.

REVIEWERS' COMMENTS

Reviewer #1 (Remarks to the Author):

The authors have done a meticulous and satisfactory job of addressing my comments, as well as those of the other reviewers, and I commend them for doing so. I would like the authors to tackle the following two minor matters, after which the manuscript ought to be suitable for publication.

We are grateful for the thorough reviews which greatly improved the manuscript.

1. As the authors would have noticed, the paper by Hao et al. (2022) appeared in PNAS subsequent to their initial submission, which contains similar (but not identical) results. Therefore, I would like to see a moderately expanded discussion (in the Introduction) of how the framework employed by the authors diverges from and/or overlaps with Hao et al. (2022). This discussion is important in order to highlight the novel aspects of the authors' approach, as well as to underscore how the two studies broadly agree with each other.

We have updated the manuscript to highlight the complimentary Enceladus-focused paper you mentioned by contextualizing their results and focus on equilibrium conditions with the Enceladus ocean.

“For the specific case of Enceladus, our focus on the range of hydrothermal conditions and potential dynamic controls on ocean phosphate abundance provides a distinct and complementary perspective in relation to Hao and co-workers' focus on equilibrium with respect to bulk ocean conditions.”

2. With regard to lines 324-326, there are certain studies that have obtained cell densities greater than 10,000 cells/mL. One recent example is Affholder et al. (2022) - which incorporates both nutrient and energetic constraints - and some additional studies and simple calculations are provided in Chapter 7.5 of Lingam & Loeb (2021).

<https://iopscience.iop.org/article/10.3847/PSJ/aca275/meta>

<https://www.hup.harvard.edu/catalog.php?isbn=9780674987579>

Hence, I would recommend modifying lines 324-326 slightly, and citing the aforementioned reference(s).

We have updated as suggest.

Reviewer #2 (Remarks to the Author):

The authors have addressed all of my concerns. The manuscript is ready for publication.

We are grateful for the thorough reviews which greatly improved the manuscript.